# Novel 1,4-Dihydropyridine Derivatives as Mineralocorticoid Receptor Antagonists

**DOI:** 10.3390/ijms24032439

**Published:** 2023-01-26

**Authors:** Felipe Luis Pérez-Gordillo, Natalia Serrano-Morillas, Luz Marina Acosta-García, María Teresa Aranda, Daniela Passeri, Roberto Pellicciari, María Jesús Pérez de Vega, Rosario González-Muñiz, Diego Alvarez de la Rosa, Mercedes Martín-Martínez

**Affiliations:** 1Instituto de Química Médica (IQM-CSIC), Juan de la Cierva, 3, 28006 Madrid, Spain; 2Departamento de Ciencias Médicas Básicas and Instituto de Tecnologías Biomédicas, Universidad de La Laguna, 38200 La Laguna, Spain; 3Tes Pharma, Via P. Togliatti 20, 06073 Corciano, Italy

**Keywords:** MR, nuclear receptor, NR3C2, aldosterone, DHP, antagonist, docking, molecular dynamics

## Abstract

The mineralocorticoid receptor (MR) belongs to the steroid receptor subfamily of nuclear receptors. MR is a transcription factor key in regulating blood pressure and mineral homeostasis. In addition, it plays an important role in a broad range of biological and pathological conditions, greatly expanding its interest as a pharmacological target. Non-steroidal MR antagonists (MRAs) are of particular interest to avoid side effects and achieve tissue-specific modulation of the receptor. The 1,4-dihydropyridine (1,4-DHP) ring has been identified as an appropriate scaffold to develop non-steroidal MRAs. We report the identification of a novel series of 1,4-DHP that has been guided by structure-based drug design, focusing on the less explored DHP position 2. Interestingly, substituents at this position might interfere with MR helix H12 disposition, which is essential for the recruitment of co-regulators. Several of the newly synthesized 1,4-DHPs show interesting properties as MRAs and have a good selectivity profile. These 1,4-DHPs promote MR nuclear translocation with less efficiency than the natural agonist aldosterone, which explains, at least in part, its antagonist character. Molecular dynamic studies are suggestive of several derivatives interfering with the disposition of H12 in the agonist-associated conformation, and thus, they might stabilize an MR conformation unable to recruit co-activators.

## 1. Introduction

The mineralocorticoid receptor (MR) is a ligand-activated transcription factor mediating the biological effects of aldosterone, a key factor in regulating mineral homeostasis, acid-base balance, and blood pressure [1,2,3]. In addition, MR can act as a high-affinity glucocorticoid receptor [1,4]. The main element regulating ligand access to the receptor is the enzyme 11-ß-hydroxysteroid dehydrogenase type 2 (11ßHSD2), which catalyzes the conversion of glucocorticoids into biologically inactive metabolites [5]. Co-expression of 11ßHSD2 with MR will therefore determine the main hormone activating the receptor. MR antagonists (MRAs), chiefly spironolactone, have long been used to treat primary aldosteronism and also in other clinical situations where a decrease in the extracellular volume is important, such as essential hypertension or edema originating from cirrhosis or congestive heart failure [6]. The recognition that MR has a broad tissue expression and can also be activated by glucocorticoids prompted research that ultimately showed that inappropriate activation of this receptor promotes tissue inflammation, oxidative stress, and fibrosis, suggesting possible additional uses of MRAs. For instance, excessive MR activity promotes deleterious cardiovascular and renal remodeling [7,8], both directly and through the modulation of innate and adaptive immune cells [9]. MR has also been implicated in the development of metabolic abnormalities [10,11], inflammatory skin diseases [12], or retinal and choroidal pathology [13]. In addition, MR plays a key role in stress adaptation in the brain, and its dysregulation affects cognition and behavior [14].

As a result of extensive research demonstrating the key pathophysiological roles of MR and of pre-clinical and clinical research showing the usefulness of pharmacological inhibition of this pathway, the use of MRAs has expanded considerably [15]. MR inhibition is nowadays an essential tool in treating chronic heart failure [16,17,18], and additional indications for the use of MRAs have been approved or are currently being investigated [19]. First-generation MRAs such as spironolactone suffer from significant drawbacks. Inhibition of MR decreases kidney K^+^ secretion, potentially leading to hyperkalemia [6,16,20]. In addition, spironolactone has a similar structure to progesterone and acts as a progesterone receptor (PR) agonist and androgen receptor (AR) antagonist, therefore promoting sexual side effects including gynecomastia and impotence in men and disruption of the menstrual cycle in women. The lack of MR-selectivity was solved by the development of a second-generation MR antagonist, 9–11α-epoxymexrenone (eplerenone), which displays a negligible affinity for PR or AR [21]. However, eplerenone synthesis is technically challenging, and its potency is lower than spironolactone. In addition, eplerenone use limits sexual side effects but not hyperkalemia [17], thus limiting its usefulness. This, in turn, challenged the field to develop new potent and selective MRAs, ideally with tissue-specific modulatory properties to avoid potential complications derived from hyperkalemia. To achieve these characteristics, and taking into account the early example of cyproheptadine modulatory effects on MR [22], the focus of MR pharmacology has shifted to the search for non-steroidal compounds [6,23,24,25].

The 1,4-dihydropyridine (1,4-DHP) scaffold, **1**, is a suitable skeleton in the search for non-steroidal MR ligands. Felodipine, **2** (Figure 1), a Ca^2+^ channel blocker, was the first DHP identified as an MR antagonist through the screening of Pfizer’s in-house collection [26]. Therefore, several pharmaceutical companies, namely Pfizer, Bayer, and Merck, together with other research groups, have pursued this scaffold in the search for potent and selective MR ligands [27]. The data collected in these studies have provided insights into the structure-activity relationship of this family of derivatives. Thus, at position 4, several heteroaromatic rings are suitable; in particular, the 2-chloro-4-fluorphenyl moiety has been thoroughly explored [28,29]. In addition, the stereochemistry at DHP C4 is of relevance, with one of the isomers showing higher affinity, even though the majority of these derivatives have been biologically assayed as enantiomeric mixtures [30]. At positions 3 and 5 of the DHP ring, CN and esters are appropriate for MR interaction [28]. It is worth mentioning that position 2 of the 1,4-DHP has been less explored. Nevertheless, a methylene linker tethering five-membered nitrogen heterocycles, such as triazol, tetrazol, or imidazole (**3**, Figure 1), resulted in several high-affinity ligands [30].

Here, we report on the design of a virtual 1,4-DHP-focused library that has been filtered by molecular docking studies. These theoretical calculations led to the selection of a subset of derivatives, differently substituted at C2, which have then been synthesized. The biological evaluation showed that some of these DHP derivatives are able to inhibit MR, showing selectivity against other nuclear receptors.

## 2. Results and Discussion

### 2.1. Selection of New 1,4-DHPs by Molecular Docking Studies on Virtual Focused Libraries

Based on the importance of the 1,4-DHP as central scaffold in the search for MR ligands, we aimed to get deeper insight into the structural requirements of the substituent at position 2. To this end, we designed a targeted library of 1,4-DHP. Previous studies identified as suitable moieties for MR interaction a carboxymethyl, a 2-chloro-4-fluorophenyl ring, a nitrile and a methyl at positions 3, 4, 5, and 6 of the 1,4-DHP ring, respectively [25]. Thus, on the base of this pattern of substituents on the 1,4-DHP scaffold, our first focused library explored substituents of different nature at C2. To get molecular diversity at this position, we took advantage of amino acid side chains as the source of diversity. As shown below, the synthetic scheme allows the use of β-keto esters derived from amino acids as reactants. Normally, in the preparation of the 1,4-DHP skeleton, a new chiral center is created at C4, and, hence, achiral reactants will lead to DHPs as racemic mixtures. However, amino acids, except Gly, provide a second chiral center, leading to diastereomeric mixtures that might allow the isolation of both 1,4-DHP isomers. Therefore, for the generation of the 1,4-DHP focused library, proteinogenic amino acids of different nature were selected, namely Gly, Ala, Val, Phe, Tyr, Asn, Gln, Asp, Glu, and Lys, and both L and D configurations were considered (Figure 2, R^1^ correspond to the amino acid side chain, and Appendix A). This virtual library was filtered by induce fit docking studies (IFD) to select those derivatives that theoretically were more satisfactory for the MR binding site. Regarding the stereochemistry at C-4, for the IFD studies, only (R)-1,4-DHPs were analyzed based on previous studies showing that this configuration led to higher affinity ligands compared to the S-counterpart [30]. Subsequently, this library was filtered through induce fit docking studies (IFD) between each 1,4-DHPs and the ligand binding domain (LBD) of MR (PDB code 2A3I), selecting the orthosteric binding site as the binding pocket. In order to carry out these simulations, the IFD protocol within the Schrödinger Suite of programs was used [31,32]. This protocol allows the flexibility of the ligand and the receptor amino acid side chains in a window of 5 Å from the ligand.

The analysis of the molecular docking poses showed a similar binding mode as that previously reported for other 1,4-DHPs [28,30,33]. According to the IFD data, the phenyl ring shares the binding pocket of the steroid A ring, with the fluorine atom pointing toward Arg817 and Gln776 of MR and acting as a proton acceptor (Figure 2B). The 1-NH group, a key moiety within this series, is involved in hydrogen bonding with the Asn770 side chain, an interaction that in steroid ligands is formed through the C21 hydroxyl group. Regarding our main point of interest, the substituent at position 2, it is interesting to note that small L amino acids side chains, as those of Ala (R^1^ = CH_3_) or Val [R^1^ = CH(CH_3_)_2_] (Figure 2), are partially superimposing the steroid ring D. In addition, their α–NH_2_ groups are within a suitable distance to establish an extra hydrogen bond with the CO group of the Asn770 side chain (Figure 2). The latter hydrogen bond is likewise observed for the Gly-containing DHP (R^1^ = H), which fits nicely within the orthosteric site. On the other hand, bulkier amino acids led to steric clashes. Although D-amino acids with small substituents at R^1^ might also be accommodated in the binding pocket, the amino acid side chains did not superimpose with the steroid ring.

A close inspection of the IFD pose of these 1,4-DHPs showed that the α-NH_2_ moiety at position 2 is close to the T945 residue of MR (Figure 2B). Therefore, it may be envisaged that a hydrogen bond acceptor, as a carbonyl group directly attached to this α-NH_2_ or through a linker, might be able to participate in a hydrogen bond with T945. Moreover, the substituent at position 2 points towards the loop preceding H12. According to this observation, it was also of interest to explore large substituents to ascertain whether they might modify the disposition of MR helix H12, a key helix for coactivators interaction, and thus alter the recruitment of coactivators. To this end, a second directed virtual library was generated with different substituents attached to the α-NH_2_ of the simplest member of the first focused library, the Gly containing DHP (R^1^ = H) (Figure 2A right and Appendix A). As R^2^ groups, COCH_3_, COCH_2_-OH, CO(CH_2_)_2_-OH, COCH_2_-COOH, and CO(CH_2_)_2_-COOH, together with amino acids Asp and Glu, were explored. The analysis of this second library showed that derivatives shown in Figure 2C fit nicely within the orthosteric site, and in agreement with the hypothesis, a CO group within the substituent at 1,4-DHP position 2 was either able to establish a hydrogen bond with T945, or situate in close proximity. Additionally, an extra hydrogen with Asn770 sidechain was also observed for several derivatives; such is the case for the DHP with R^2^ = CH_2_-COOH depicted in Figure 2C. On the other hand, for the Glu-containing DHP, there is a hydrogen bond between Glu α-NH_2_ and the side chain oxygen of T945 besides the Glu chiral center might allow the separation of both DHP diastereoisomers.

On the whole, from the theoretical studies, six 1,4-DHP were selected for synthesis based on IFD score, hydrogen bonds, and visual inspection. These compounds correspond to the Gly, Ala and Val containing DHP (R^1^ = H, CH_3_ and CH(CH_3_)_2_, respectively, Figure 2A) from the 1st library, together with three derivatives with a substituent attached to Gly α-NH_2_ selected from the 2nd library (Figure 2A right).

### 2.2. Synthesis of 1,4-DHPs

The preparation of 1,4-DHP through the Hantzsch reaction is well documented. However, the classical method, one-pot three-component synthesis, is not the most suitable to prepare asymmetric 1,4-DHPs, as those in our libraries. In this sense, a modification of the classical method, based on a two steps synthetic route, known as the two-component Hantzsch synthesis, is preferred for the synthesis of asymmetric 1,4-DHPs [34]. In addition, MW-assisted synthesis has been shown to increase the yields while decreasing the time of the Hantszch reaction, especially when sterically hindered aldehydes are involved, as the one needed for the preparation of our selected DHPs [34,35]. Therefore, a modified Hantszch synthesis was applied, consisting of a two step-one vial microwave-assisted reaction. Following this procedure, we set to prepare the simple derivative from the 1^st^ focused library, namely the Gly containing DHP **18** (Figure 1). Toward this goal, the first step was the condensation between the glycine-derived β-keto ester **4** and 2-chloro-4-fluorbenzaldehyde, to generate the Knoevenagel adduct, which was not isolated. Then, 3-aminocrotonitrile was added to the reaction mixture to obtain intermediate 1,4-DHP **11** as a racemic mixture. The analysis of the reaction crude leading to **11** by HLPC-MS showed the existence of several signals with the same mass. We hypothesized that they might correspond to several DHP isomers. By analyzing the evolution of this reaction by ^1^H-RMN in CDCl_3_, we realized that the initial mixture of isomers was slowly converted in solution into the more stable 1,4-DHP. Thus, based on this finding, we incorporated a third step to our synthetic route consisting in heating the crude residue at 60 °C for three days in CHCl_3_. Finally, the selected DHP **18** was prepared by removal of the benzyloxycarbonyl group from **11** by catalytic hydrogenation in the presence of 10% Pd/C, which led quantitatively to compound **18** (Figure 1). However, derivative **18** was found to slowly decompose over time, which, together with preliminary biological data that indicated that compound **11** was able to bind to the MR, motivated that the benzyloxycarbonyl group was not removed in the synthesis of the other designed DHPs.

The IFD studies on the 1st library also showed an interest in small hydrophobic substituent at the 2-methylene of **18**, in particular R^1^ = CH_3_ or R^1^ = CH(CH_3_)_2_ (Figure 2A). As indicated in Figure 1, these compounds were obtained with moderate yields following the same two-step one-pot procedure used in the synthesis of **11**, starting with β-ketoesters derived from L-Ala, **5** or L-Val, **6**. It is worth pointing out that these amino acids β-ketoesters provide a second chiral center to the 1,4-DHPs. Consequently, **12ab** and **13ab** were obtained as diastereomeric mixtures (notations **a** and **b** are used to indicate each diastereoisomer). However, only **12ab** (1:1 diastereomeric mixture) could be resolved by semipreparative HPLC.

In addition to the compounds selected from the libraries, it would be of interest to explore substituents that have led to improvement in affinity in other 1,4-DHP derivatives [28,29]. Therefore, on the base of **11**, bulkier esters were incorporated at position 3, in particular, ethyl (**14**) and tert-butyl (**15**), a 4-cyano-2-methoxi-phenyl moiety at position 4 (**16**), and a carboxamide at position 5 (**17**). As shown in Figure 1, these 1,4-DHPs were obtained in low yield by the two component Hantzsch synthesis starting from the corresponding Gly β-ketoesters **7–10**.

Although the synthetic route has allowed the preparation of the selected DHP, in general, they were obtained in low to moderate yields, and the methodology requires heating in CHCl_3_ for 1–3 days. The need for further amounts of DHP **18** to prepare derivatives from the 2nd library motivates further studies aimed at optimizing the synthetic methodology. As the solvent CDCl_3_ usually has HCl traces, we envisaged that acid catalysis might improve the interconversion of DHP isomers. Consequently, the preparation of **11** (precursor of **18**) using CH_3_CN/HCl in the last step led, after just 30 min, to this 1,4-DHP with a considerable increase in the yield, from 28% to 55% (Figure 1). To analyze whether this yield improvement might be extended to other DHPs, we explored the use of this acid catalysis for the preparation of **12**, **13**, and **16**. As shown in Figure 1, there was always an increase in the yield varying from a 15% to a 39% improvement. These data show that the synthetic route can benefit from adding acid catalysis in the last step. To the best of our knowledge, this methodology is not described yet and could be of more general application for the preparation of other 1,4-DHP, with improved yield and substantial decrease in the reaction time.

The 2nd focused library was based on the Gly containing DHP **18**, and the selected 1,4-DHP could be easily obtained from it through an acylation reaction, as shown in Figure 2. The reaction of **18** with acetyl chloride, in the presence of propylene oxide, led to 1,4-DHP **19**. On the other hand, the preparation of 1,4-DHP **20** and **22** required the coupling between **18** and succinic or glutamic acid derivatives, respectively. These acylation reactions were performed in soft conditions, using PyBOP as a coupling agent and TEA as the base. As expected, **22ab** was obtained as a mixture of diastereoisomers, but nonetheless, the efforts to isolate the isomers were unsuccessful. However, after deprotection of the benzyloxycarbonyl group of **22ab** by catalytic hydrogenation in the presence of 10% Pd/C, the 1:1 diastereomeric mixture could be resolved, leading to pure isomers **23a** and **23b**. On the other hand, the deprotection of the tert-butoxy carbonyl group of **20** by treatment with TFA led to derivative **21** in quantitative yield.

### 2.3. Biological Evaluation

The synthesized 1,4-DHP derivatives were initially evaluated for their ability to displace [^3^H]-aldosterone from human MR (hMR) in a cell-based assay. In Figure 3, it can be observed that all of the 1,4-DHP tested, except **23b**, were able to interact with MR, thus validating the theoretical approach used to design these new compounds.

We next took advantage of the [^3^H]-aldosterone assay to calculate inhibitory constants (Ki) for a selection of 1,4-DHP derivatives. To that end, we used a constant concentration of [^3^H]-aldosterone (1 nM) and increasing concentrations of each derivative. Curves obtained with this procedure are shown in Figure 4, and a summary of the results is presented in Table 1.

In the first directed library, we selected two derivatives with small substituents at R^1^, **12**, and **13**, together with **11**, as bulkier ones seem to lead to steric clashes. From the comparison of these derivatives, it has to be taken into account that compound **11** is a mixture of enantiomers, **13ab** of diastereoisomers, whereas for derivative **12**, both diastereoisomers could be isolated and assayed separately. The analysis of the biological data indicated that even a small substituent, such as the methyl group in **12a**, led to a slight increase in Ki compared with **11** (Table 1). This data is suggestive of a narrow pocket around this substituent, leading to a decrease in affinity as the size of R^1^ increases. On the other hand, at position R^2^ several ester derivatives have been prepared (**11**, **14**, and **15**), and biological assay indicates that the derivative with the ethyl ester, **14**, is the one with better affinity (1.4 decrease in Ki in comparison with methyl ester DHP **11**), while a further increase in size, as in the ^t^Bu compound **15**, reduces affinity. Regarding the DHP CN group, its replacement by an amide, as in derivative **17**, results in a 3.2 increase in Ki, whereas the substitution of the 2-chloro-4-fluorbenzyl moiety at DHP C4 in **11** by a 4-ciano-2-methoxyphenyl as in **16** led to an increase in affinity. It is worth pointing out that this phenyl ring is equivalent to one of the substituents of finerenone [29]. Overall, for this set of compounds, the best substituents found are an ethyl ester at R^2^, 4-ciano-2-methoxyphenyl at DHP position 4, and no substituent at R^1^.

Derivatives from the second directed library showed that at position R^6^ the replacement of the Cbz group for the smaller COCH_3_ in compound **19** led to a 3.3 increase in affinity, while bulkier substituents, as in **20** or **23**, results in decreased affinities.

It is interesting to point out, that as expected, when both 1,4-DHP diastereoisomers could be isolated, one of them has better affinity, as **12a** vs. **12b** or **23a** vs. **23b**, in agreement with previous studies for other 1,4-DHP derivatives. Although we have not been able to establish the chirality of the 1,4-DHP C4 center unequivocally, based on previous biological data, it would be expected that the more active derivatives were those with an R configuration at DHP C4 [30].

The next step in the biological evaluation of the new derivatives was to test whether they have effects as agonists or competitive antagonists on MR. To that end, we performed gene reporter transactivation assays in transiently transfected COS-7 cells. In each assay, we transfected MR and stimulated it or not with saturating concentrations of aldosterone (10 nM) for 24 h in the presence or absence of the test ligands. Figure 5A shows that in the absence of aldosterone, none of the derivatives produce any significant increase on basal MR activity, ruling them out as potential agonists, at least at the concentration tested (5 μM). When cells were stimulated with aldosterone, eplerenone produced the expected inhibitory effect, decreasing luciferase activity to an average of 38% of maximal induction, which is similar to previously described results under comparable conditions [42]. Derivatives **16** and **20** produced statistically significant MR inhibition, reducing activity to 57% and 48% of the maximal activity. Based on these results, it appears that the inhibitory potency of the derivatives is lower than what would be expected from their calculated Ki. In fact, this is in line with what has been described with other known competitive MRAs. For instance, eplerenone Ki is approximately 200 nM [43], while its IC_50_ calculated using a luciferase-based transactivation assay similar to the one used in this study is 990 nM [33]. Other MRAs, such as spironolactone or finerenone, also show significantly higher IC_50_ values than values measuring affinity (spironolactone: Ki = 2 nM [44], IC_50_ = 24 nM [33]; finerenone: Kd = 1.5 nM, IC_50_ = 58 nM [45]). In general, the higher IC_50_ values, when compared with Ki or Kd may be partially explained by the fast off-rate kinetics of MRAs, including finerenone, when compared to aldosterone [45].

MR activation involves an initial step of ligand-induced receptor translocation from the cytosol to the nucleus. The agonist aldosterone triggers complete MR nuclear translocation within one hour (Figure 6). Antagonists such as finerenone or spironolactone also induce nuclear translocation, although partial and with slower kinetics [45]. To further characterize the new 1,4-DHP derivatives, we performed nuclear translocation assays in the absence of aldosterone with compounds **11–17**, **19**, and **20**. With the exception of compound **17**, all promoted nuclear translocation with slower kinetics compared to aldosterone (Figure 6 and Appendix A). Accordingly, even at 24 h after the application of these 1,4-DHP derivatives, there is still a fraction of eGFP-MR in the cytoplasmic compartment. Comparing derivatives in terms of affinity shows interesting differences in translocation kinetics. For instance, at 24 h, compounds **16** and **19**, which have comparable affinities (262 and 152 nM, respectively), show differential nuclear localization, with **19** inducing higher nuclear accumulation compared to **16**. Compound **20**, with significantly lower Ki (858 nM) than **16**, shows a higher fraction of MR at the nucleus after 24 h. Given that compounds **16** and **20** both inhibit aldosterone-induced MR activity, our data suggest that the novel 1,4-DHP derivatives will likely decrease MR activity through a combination of mechanisms, including partial competition of MR nuclear translocation.

An issue with MR ligands is the potential lack of selectivity against other nuclear receptors [6,42]. To ascertain if our derivatives were selective for MR over other NRs, we assayed **11** and **19** across a panel of human nuclear receptors using AlphaScreen assays. The receptors included in these studies were LXRα, LXRβ, PPARα, PPARδ, PPARγ, PXR, ERα, GR, PR, and TRβ. In the agonist mode, they only interact with the promiscuous receptor PXR: **11**, EC_50_ 0.95 μM, 90% efficacy; **19,** EC_50_ 13 μM, 80% efficacy. In the antagonist mode, they only show a weak effect on the PR receptor (**11**, IC_50_ 23 μM, 55% inhibition; **19** IC_50_ 20 μM, 40% inhibition). Overall, both compounds displayed a good selectivity profile.

To further test selected compounds on the activity of related nuclear receptors, we performed reporter gene transactivation assays with GR and AR. Derivatives **11**, **16**, **19**, **20**, and **23a** were added to COS-7 cells transiently transfected with plasmids encoding the human GR and AR receptors. Cells were treated or not with known agonists (cortisol or testosterone, respectively) in the presence or absence of 1,4-DHP derivatives. None of the compounds tested showed agonist activity on either receptor in this assay (Appendix A). When cells were stimulated with the respective agonists, only derivative **16** showed a minor inhibitory effect on AR, decreasing activation by 15% (Appendix A).

### 2.4. Molecular Dynamic Simulations

To get further insights into the binding mode of these DHP, molecular dynamics (MD) simulations were carried out for derivatives **11**, **19**, and **20**. These simulations were done using the AMBER ff19SB force field for the protein and GAFF for the ligand in an explicit water environment for 500 ns. The results of the MD simulations have provided further information regarding key interactions between these ligands and the MR. In this sense, the DHP NH of derivatives **19** and **20** is participating in a hydrogen bond with MR N770 in over 90% of the frames during the 500 ns simulation (Figure 7A,B, Appendix A), in agreement with the importance highlighted for this NH [28]. On the other hand, both derivatives, **19** and **20**, establish an extra hydrogen bond with the N770 side-chain in over 53% of the frames, either through the NH or CO at position 2 of the DHP ring, respectively. Additionally, a hydrogen bond with the hydroxyl of the S810 side chain is observed for **19** and **20** via the CN group (around 48%). This latter hydrogen bond might be behind the selectivity of these derivatives, as this residue corresponds to Met in AR, GR, and PR [28]. Additionally, Ala773, a residue unique to MR (a Gly residue in AR, GR, and PR) is surrounded by the substituents at C4, C5, and C6 of the DHP ring, which might also contribute to selectivity. This is in agreement with previous publications regarding DHP selectivity [28]. Finally, the fluorophenyl ring is oriented toward the Q776/R817 cavity. It is interesting to note that **20** has an intramolecular hydrogen bond that stabilizes a conformation in which the tert-butyl ester is partially overlapping the steroid D ring (Figure 7E). However, the lower affinity compared with **19** is likely indicating steric clashes. The analysis of helix H12 showed that these derivatives are likely able to accommodate within the binding pocket without affecting the disposition of this helix. Thus, their antagonist character might be due to the lack of hydrogen bonds with several MR residues, among them T945, which, similarly to what has been hypothesized for spironolactone, might debilitate the hydrogen bond network [46].

On the other hand, IFD studies for derivative **11,** using the same protocol applied for filtering the focused libraries, led to a variety of poses with few interactions with MR. As mentioned previously, the substituent at position 2 was oriented toward the link preceding helix H12; thus we performed additional IFD studies removing both H12 and the preceding loop. In these conditions, suitable poses were obtained, which situated the DHP ring in a similar position as that observed for 1,4-DHP **19**. Subsequent MD simulations showed an equilibrium between two rather similar binding modes (Figure 7F, and Appendix A). In one of them, the N770 side chain CO group is participating in two hydrogen bonds with both NH moieties of **11** (Figure 7E). Besides the N770 side chain NH_2_ group is involved in a hydrogen bond with the sp3 oxygen of the benzyloxicarbonyl group, a conformation that was observed in around 56% of the frames. On the other hand, there are about 33% of the frames that are not establishing these hydrogen bonds, although the 3D ligand pose is quite similar to the previous one (Figure 3F). In this latter conformation, there is a movement of the N770 sidechain, allowing it to establish a stacked amide π interaction between the N770 sidechain amide, and the benzyloxycarbonyl phenyl ring of **11**. In both poses, the benzyloxycarbonyl group would partially overlap MR residues V954 and F956 within the loop preceding helix H12, as well as L960 on H12, provided that they had not been removed previously to the simulation. Therefore, it can be hypothesized that derivative **11** is a bulky passive antagonist, which stabilized the MR in a conformation unable to recruit co-regulators [33].

## 3. Conclusions

Molecular docking studies have guided the selection of a subset of 1,4-DHPs differently substituted at C2 that might either provide additional interactions with MR or modify the disposition of the MR helix H12. Although initially prepared with low to moderate yields, the incorporation of an acid-catalyzed step aid in the interconversion of DHP regioisomers improves the yields, and decreases reaction time. Therefore, this methodology might be of general interest in the preparation of other 1,4-DHPs.

Biological evaluation of the synthesized 1,4-DHPs showed that they behave as MR antagonists, with several compounds able to bind to MR with submicromolar affinity. Biological and MD studies shed light on the structure-activity relationship regarding substituents at DHP position 2. Selectivity against other nuclear receptors was observed for several DHP derivatives, which might be attributed to the substituent at C4, C5, and C6 of the 1,4-DHP ring and/or hydrogen bonds with S810. These antagonists, similar to eplerenone or finerenone, are able to promote a partial nuclear translocation of MR. Finally, MD simulations are suggestive of two different modes of antagonism, either through debilitating the hydrogen bond network, as in **19** and **20**, and/or acting as a bulky antagonist able to modify the MR helix H12 disposition, as in **11**. Whether these potential behaviors might influence the pattern of co-regulator recruitment or might lead to a different biological effect remains to be investigated.

## 4. Materials and Methods

### 4.1. Molecular Modelling Studies

#### 4.1.1. Induced Fit Docking Studies

MR LBD was taken from the PBD structure code 2A3I, and prepared with the Protein Preparation Wizard tool within the Schrödinger suite of programs (Protein Preparation Wizard 2015-4; Epik version 2.4, S., LLC, New York, NY, USA, 2015; Impact version 5.9, Schrödinger, LLC, New York, NY, USA, 2015; Prime version 3.2, Schrödinger, LLC, New York, NY, USA, 2015; [47]). The ligands were constructed using Maestro 2015-4 (Maestro, Schrödinger, LLC, New York, NY, USA, 2015) and prepared with the LigPrep application in the Schrödinger suite of programs [48]. Then, IFD studies were carried out with the IFD protocol 2015-4 within the Schrödinger suite of programs [31,32]. The docking site was selected as a cubic box centered on the ligand. The IFD protocol allowed the flexibility of side chains of the residues within 5 Å from the ligand. Thus, there is an initial Glide docking, in which the van der Waals radii (rdW) of both protein and ligand are scaled by 0.5 to reduce steric clashes. Then, the side chains of residues within 5 Å from the ligand were optimized with Prime (Schrödinger, LLC, New York, NY, USA, 2015; [49,50]). Finally, the ligands were re-docked with Glide, using the default rdW radii, into the new MR LBD conformation generated. Poses were ranked using IFD score. The selection was based on the best-ranked poses and on visual inspection. For visualization and creating figures the PyMOL Molecular Graphic System v2.3.3 (Schrödinger Inc., München, Germany) was used.

#### 4.1.2. Molecular Dynamic Studies

The best IFD poses for 1,4-DHPs **11**, **19**, and **20** were selected to get further insights into their binding modes by molecular dynamics (MD) studies using Amber 20 and the FF19SB force field [51]. In this way, the flexibility of the whole protein is taken into account. Each MR LBD-1,4-DHPs complex was solvated in an octahedric box using TIP3P water molecules with each site at least 10 Å from any complex atom. Periodic boundary conditions were used. First, a steepest descent minimization for 30,000 steps was carried out, initially with a positional restraint weight of 5.0 Kcal mol^−1^ A^−2^, restraints were progressively weakened and removed in the last 10,000 steps. Subsequently, the system was equilibrated over 125 ps, initially using a positional restraint weight of 5.0 Kcal mol^−1^ A^−2^ and an integration time step of 1.0 fs. For 100 ps, water was equilibrated, while the solutes where restrained. Restraints were progressively weakened untill they were removed in the last 25 ps. The temperature was maintained using Langevin dynamics at 300 K. Once the system was equilibrated, a production simulation of 500 ns was run under the NPT ensemble, with an integration time step of 2.0 fs and the SHAKE algorithm to constrain hydrogen bonds. The particle mesh Ewald method was applied to calculate electrostatic interaction. For visualization and creating figures, the PyMOL Molecular Graphic System v2.3.3 (Schrödinger Inc., München, Germany) was used.

### 4.2. Chemistry

#### 4.2.1. Synthesis of Dihydropiridines

A solution of the corresponding aldehyde (0.2 mmol) in 0.5 mL of a mixture of isopropanol, 1% of piperidine, and 1% of acetic acid was treated with the corresponding β-keto ester (0.2 mmol). After 1–5 min of stirring at 40 °C, the corresponding aminocrotonate (0.22 mmol) was added, and the reaction was irradiated with a microwave at 100 °C for 3–15 min. Subsequently, the solvent was removed under reduced pressure, and the residue was treated according to method A or B. Method A: The resulting residue was dissolved in CHCl_3_ (1 mL) and heated in a sealed tube at 60 °C for 1–3 days. After evaporation of the solvent to dryness, the resulting reaction mixture was purified in a silica gel column chromatography, using the eluent systems specified in each case. Method B: The resulting residue was dissolved in 1 mL of acetonitrile containing 1% of concentrated HCl and stirred at room temperature for 30 min. After evaporation of the solvent to dryness, the resulting crude was purified in a silica gel column chromatography or through semipreparative HPLC, using the eluent systems specified in each case.

Methyl 2-[(benzyloxycarbonyl)amino]methyl-4-(2-chloro-4-fluoro)phenyl-5-cyano-6-methyl-1,4-dihydropyridine-3-carboxylate (11). From methyl-4-[(benzyloxycarbonyl)amino]-3-oxo-butanoate, 2-chloro-4-fluorobenzaldehyde and 3-aminocrotonate. Method A: 1 min (40 °C), 5 min (100 °C), 1 day (CHCl_3_). Yield. 28%. Method B: Yield 55%. Syrup. Eluent: Hexane:EtOAc (2:1). HPLC-MS: Gradient from 30 to 95% ACN/H_2_O (0.05% TFA) in 10 min, t_R_: 8.11, (*m*/*z*: 470.17 M+H^+^). HRMS (ESI pos) *m*/*z* Calculated C_24_H_21_ClFN_3_O_4_ 469.12046, found 469.12201 (3.31 ppm). **^1^H NMR** (400 MHz, CDCl_3_): δ 7.43 (s, 1H, 1-H), 7.41–7.32 (m, 5H, Cbz Ar), 7.19 (dd, J = 8.7, 6.0 Hz, 1H, Ph 6-H), 7.08 (dd, J = 8.6, 2.6 Hz, 1H, Ph 3-H), 6.90 (ddd, J = 8.6, 8.3, 2.6 Hz, 1H, Ph 5-H), 5.75 (t, J = 5.5 Hz, 1H, NHCO), 5.15 (s, 3H, 4-H, Cbz CH_2_), 4.37 (dd, J = 15.0, 6.6 Hz, 1H, 2-CH_2_), 4.29 (dd, J = 15.0, 6.7 Hz, 1H, 2-CH_2_), 3.54 (s, 3H, OCH_3_), 2.04 (s, 3H, 6-CH_3_). **^13^C NMR** (100 MHz, CDCl_3_): δ 167.2 (3-COO), 161.4 (d, J = 249.5 Hz, Ph 4-C), 158.5 (Cbz CO), 146.6 (2-C), 145.5 (6-C), 139.2 (d, J = 3.7 Hz, Ph 1-C), 136.0 (Cbz, Ph C), 133.2 (d, J = 10.4 Hz, Ph 2-C), 131.5 (d, J = 8.8 Hz, Ph 6-C), 128.8, 128.6, 128.2 (Cbz, Ph CH), 118.9 (CN), 116.9 (d, J = 24.5 Hz, Ph 3-C), 114.9 (d, J = 21.1 Hz, Ph 5-C), 101.6 (3-C), 86.2 (5-C), 67.7 (Cbz OCH_2_), 51.6 (OCH_3_), 41.2 (2-CH_2_), 37.5 (4-C), 18.4 (6-CH_3_).

Ethyl-2-[(benzyloxycarbonyl)aminomethyl]-4-(2-chloro-4-fluoro)phenyl-5-cyano-6-methyl-1,4-dihydropyridine-3-carboxylate (14). From ethyl 4-[(benzyloxycarbonyl)amino]-3-oxo-butanoate, 2-chloro-4-fluorobenzaldehyde and 3-aminocrotonate. Method A: 5 min (40 °C), 15 min (100 °C), 3 days (CHCl_3_). Yield: 26%. Syrup. Eluent: Hexane:EtOAc (2:1). HPLC-MS: Gradient from 30 to 95% ACN/H_2_O (0.05% TFA) in 10 min, t_R_: 8.65, (*m*/*z*: 484.16 M+H^+^). HRMS (ESI pos) *m*/*z* Calculated C_25_H_23_ClFN_3_O_4_ 483.13611, found 483.13436 (−3.63 ppm). **^1^H NMR** (300 MHz, CDCl_3_): δ 7.37 (m, 6H, Cbz, 1-NH), 7.19 (dd, J = 8.6, 6.0 Hz, 1H, Ph 6-H), 7.08 (dd, J = 8.6, 2.6 Hz, 1H, Ph 3-H), 6.91 (ddd, J = 8.6, 8.3, 2.6 Hz, 1H, Ph 5-H), 5.79 (t, J = 6.8 Hz, 1H, NHCO), 5.17 (s, 1H, 4-H), 5.14 (s, 2H, Cbz CH_2_), 4.32 (m, 2H, 2-CH_2_), 3.97 (qd, J = 7.1, 2.8 Hz, 2H, Et CH_2_), 2.02 (s, 3H, 6-CH_3_), 1.07 (t, J = 7.1 Hz, 3H, Et CH_3_). **^13^C NMR** (75 MHz, CDCl_3_): δ 166.7 (3-CO), 161.4 (d, J = 249.3 Hz, Ph 4-C), 158.5 (Cbz CO), 146.5 (2-C), 145.4 (6-C), 139.3 (d, J = 3.7 Hz, Ph 1-C), 136.0 (Cbz Ph C), 133.1 (d, J = 10.4 Hz, Ph 2-C), 131.5 (d, J = 8.9 Hz, Ph 6-C), 128.8, 128.6 128.2 (Cbz Ph CH), 118.9 (CN), 116.8 (d, J = 24.6 Hz, Ph 3-C), 114.8 (d, J = 21.0 Hz, Ph 5-C), 101.8 (3-C), 86.0 (5-C), 67.7 (Cbz CH_2_), 60.6 (Et 1-C), 41.2 (2-CH_2_), 37.4 (4-C), 18.4 (6-CH_3_), 14.0 (Et 2-C).

Methyl 2-[(benzyloxycarbonyl)amino]methyl-5-cyano-4-(4-cyano-2-methoxy)phenyl-6-methyl-1,4-dihydropyridine-3-carboxylate (16). From methyl 4-[(benzyloxycarbonyl)amino]-3-oxo-butanoate, 2-chloro-4-fluorobenzaldehyde and 3-aminocrotonate. Method A: 1 min (40 °C), 4 min (100 °C) and 1 day (CHCl_3_). Yield 29%. Method B. Yield 68%. Syrup. Eluent: Hexane:EtOAc (2:3). HPLC_MS: Gradient from 30 to 95% ACN/H_2_O (0.05% TFA) in 10 min, t_R_ = 7.21, (*m*/*z*: 473.46 M+H^+^). HRMS (ESI pos) *m*/*z* Calculated C_26_H_24_N_4_O_5_ 472.17467, found 472.17422 (−0.96 ppm). **^1^H NMR** (400 MHz, CDCl_3_): δ 7.43–7.31 (m, 6H, 1-H, Cbz Ph), 7.19–7.10 (m, 2H, Ph 5-H 6-H), 7.09 (d, J = 1.4 Hz, 1H, Ph 3-H), 5.74 (t, J = 6.6 Hz, 1H, NHCO), 5.15 (d, J = 1.6 Hz, 2H, Cbz CH_2_), 5.07 (s, 1H, 4-H), 4.37 (dd, J = 15.1, 8.0 Hz, 1H, 2-CH_2_), 4.35 (dd, J = 15.1, 8.0 Hz, 1H, 2-CH_2_), 3.85 (s, 3H, Ph OCH_3_), 3.53 (s, 3H, COOCH_3_), 2.01 (s, 3H, 2-CH_3_). **^13^C NMR** (75 MHz, CDCl_3_): δ 167.2 (3-CO), 158.5 (Cbz CO), 156.7 (Ph 2-C), 147.4 (2-C), 146.2 (6-C), 138.6 (Ph C), 136.0 (Cbz Ph C), 129.7 (Ph 2-C), 128.8, 128.7, 128.3 (Cbz Ph CH), 125.1 (Ph 3-C), 119.1 (Ph CN), 119.0 (5-CN), 114.2 (Ph 3-C), 111.8 (Ph 4-C), 100.1 (3-C), 85.3 (5-C), 67.7 (Cbz CH_2_), 56.0 (Ph OCH_3_), 51.7 (COOCH_3_), 41.2 (2-CH_2_), 34.9 (4-H), 18.5 (2-CH_3_).

For experimental details, description, and characterization of derivatives **12ab**, **13ab**, **15**, **17**, **18**, and **23ab**, see Appendix A.

#### 4.2.2. Acylation of Primary Amine

Method A. To a solution of the corresponding amine (0.1 mmol) in CH_2_Cl_2_ (1 mL), it is added at 0 °C propylene oxide (1.5 mmol, 105 µL) and the corresponding acid chloride (0.15 mmol). After 18 h of stirring at room temperature, the solvent was evaporated to dryness, and the resulting residue was purified in a silica gel column chromatography using the eluent system indicated for each product. Method B. To a solution of the corresponding amine (0.1 mmol) in dry THF (2 mL) it is added the corresponding acid (0.2 mmol), PyBOP (0.2 mmol, 104 mg), and TEA (0.2 mmol, 27 µL). After 18 h of stirring at room temperature, the solvent was evaporated to dryness, and the residue was dissolved in AcOEt. The organic layer was washed with 10% citric acid, 10% NaHCO_3_, saturated NaCl, and dried with Na_2_SO_4_. The solution was filtered and concentrated. The product was purified by column chromatography using the eluent system indicated in each product.

Methyl 2-(acetamido)methyl-4-(2-chloro-4-fluoro)phenyl-5-cyano-6-methyl-1,4-dihydropyridine-3-carboxylate (19). From **18** and acetyl chloride. Method A: Yield 43%. Syrup. Eluent: Hexane:EtOAc (5:1). HPLC-MS: Gradient from 15 to 95% ACN/H_2_O (0.05% TFA) in 15 min, t_R_ = 7.45, (*m*/*z*: 378.50 M+H^+^). HRMS (ESI pos) *m*/*z* Calculated C_18_H_17_ClFN_3_O_3_ 377.09425 found 377.09292 (−3.52 ppm). **^1^H-RMN** (400 MHz, CDCl_3_): δ 7.86 (s, 1H, 1-H) 7.20 (dd, J = 8.7, 6.0 Hz, 1H, Ph 6-H), 7.08 (dd, J = 8.6, 2.7 Hz, 1H, Ph 3-H), 6.93 (ddd, J = 8.7, 7.9, 2.7 Hz, 1H, Ph 5-H), 6.61 (t, J = 6.5 Hz, 1H, NHCO), 5.14 (s, 1H, 4-H), 4.43 (dd, J = 14.3, 6.5 Hz, 1H, 2-CH_2_), 4.34 (dd, J = 14.3, 6.5 Hz, 1H, 2-CH_2_), 3.55 (s, 3H, OCH_3_), 2.06 (s, 3H, 6-CH_3_), 2.04 (s, 3H, COCH_3_). **^13^C-RMN** (100 MHz, CDCl_3_): δ 172.7, 167.5 (CO), 161.4 (d, J = 249.6 Hz, Ph 4-C), 146.8, 145.7 (2-C, 6-C), 139.3 (d, J = 3.7 Hz, Ph 1-C), 133.1 (d, J = 10.4 Hz, Ph 2-C), 131.6 (d, J = 8.9 Hz, Ph 6-C), 119.0 (CN), 116.9 (d, J = 24.4 Hz, Ph 3-C), 114.9 (d, J = 21.0 Hz Ph, 5-C), 101.5 (3-C), 85.9 (5-C), 51.6 (OCH_3_), 39.9 (2-CH_2_), 37.5 (4-C), 23.1 (COCH_3_), 18.5 (6-CH_3_).

Methyl 2-((3-(Tert-butoximalonamido)methyl-4-(2-chloro-4-fluoro)phenyl-5-cyano-6-methyl-1,4-dihydropyridine-3-carboxylate (20). From 18 and mono-tert-butyl malonate. Method B: Yield 64%. Syrup. Eluent: Hexane:EtOAc (1:1). HPLC-MS: Gradient from 15 to 95% ACN/H_2_O (0.05% TFA) in 10 min, t_R_ = 9.63, (*m*/*z*: 476.57 M+H^+^). HRMS (ESI neg) *m*/*z* Calculated C_23_H_25_ClFN_3_O_5_ 447.14668, found 447.14465 (−4.24 ppm). ^1^H NMR (400 MHz, CDCl_3_): δ 7.74 (m, 2H, NHCO, H-1), 7.22 (dd, J = 8.6, 6.1 Hz, 1H, Ph 6-H), 7.07 (dd, J = 8.6, 2.6 Hz, 1H, Ph 3-H), 6.93 (ddd, J = 8.6, 8.3, 2.6 Hz, 1H, Ph 5-H), 5.16 (s, 1H, 4-H), 4.56 (dd, J = 14.9, 6.4 Hz, 1H, 2-CH_2_), 4.50 (dd, J = 14.9, 6.5 Hz, 1H, 2-CH_2_), 3.57 (s, 3H, OCH_3_), 3.29 (s, 2H, Pr CH_2_), 2.06 (s, 3H, 6-CH_3_), 1.47 (s, 9H, ^t^Bu CH_3_). ^13^C NMR (101 MHz, CDCl_3_): δ 168.4, 168.2, 167.2 (CO), 161.4 (d, J = 249.2 Hz, Ph 4-C), 146.1, 145.6 (2-C, 6-C), 139.3 (d, J = 3.6 Hz, Ph 1-C), 133.1 (d, J = 10.4 Hz, Ph 2-C), 131.6 (d, J = 9.0 Hz, Ph 6-C), 119.0 (CN), 116.9 (d, J = 24.4 Hz, Ph 3-C), 114.9 (d, J = 21.0 Hz, Ph 5-C), 101.6 (3-C), 86.0, 83.4 (3-C, ^t^Bu C), 51.6 (OCH_3_), 42.3), 39.7 (2-CH_2_, Pr CH_2_), 37.5 (4-C), 28.1 (^t^Bu CH_3_), 18.5 (6-CH_3_).

For experimental details, description, and characterization of derivatives **22ab** and **21**, see Appendix A, respectively.

### 4.3. Biological Studies

#### 4.3.1. Plasmid Constructs and Cell Culture

Biological assays were performed using human MR (NR3C2, hMR; GenBank protein accession number, NP_000892). The hMR coding sequence was obtained originally obtained from plasmid 3750 [1] by restriction enzyme digestion with HindIII and subcloned in pcDNA3 (Invitrogen, Waltham, MA, USA) by Fagart et al. [52]. Nuclear translocation experiments used a fluorescent variant of hMR fused in its NH_2_-terminus to the enhanced green fluorescent protein (eGFP) by subcloning in peGFP-C1 (Clontech, Mountain View, CA, USA) by Ouvrard-Pascaud et al. [53]. Plasmid pcDNA3-hGR, expressing full-length human GR, has been previously described [54]. These three constructs were the kind gift of Nicolette Farman (INSERM, Paris, France). Plasmid pCMV-FLAG-hAR, expressing full-length human AR, was a gift from Elizabeth Wilson (Addgene plasmid # 89080) [55]. MR, GR, and AR transcriptional activity assays used a plasmid containing the firefly luciferase (luc) coding sequence under two copies of the basic glucocorticoid response element (GRE) cloned in pGL3-basic (Promega, Madison, WI, USA; [56]), kindly provided by Dr. Rainer Lanz (Baylor College of Medicine, Houston, TX, USA). As an internal control, transcription reporter assays included a plasmid encoding *Renilla reniformis* luciferase (ren) under the control of the SV40 promoter, cloned in pSG5 (Stratagene, San Diego, CA, USA) and kindly provided by Dr. Fátima Gebauer (Centro de Regulación Genómica, Barcelona, Spain). All plasmids were prepared from Escherichia coli cultures using a commercially available kit (NucleoBond Xtra Midi, Machenery-Nagel GmbH & Co., Düren, Germany). DNA purity and concentration were assessed by spectrophotometry using a Nanodrop 1000 system (Thermo Fischer Scientific, Waltham, MA, USA). All cell assays were performed in COS-7 cells derived from the kidney of Cercopithecus aethiops and obtained from the American Type Culture Collection (ATCC, Manassas, VA, USA; catalog number CRL-1651). These cells lack endogenous expression of MR or GR [57]. Cells were cultured and transfected as described before [57,58,59]. Briefly, cells were cultured in Dulbecco’s modified Eagle’s medium (DMEM) supplemented with 10% fetal bovine serum at 37 °C in a 5% CO_2_ atmosphere. Cells were expanded by serial passage 2–3 times weekly, up to 20 passes. For transfection, cells were seeded to provide a 70–80% density after 24 h, and plasmids were incorporated using jetPrime (Polyplus Transfection, Illkirch, France), as described [60]. Experiments were conducted 24–48 h after transfection.

#### 4.3.2. Ligand-Receptor Binding Assays

Ligand-receptor binding assays were performed as described before [60,61]. Briefly, COS-7 cells were seeded on 24-well plates and transfected with plasmid pcDNA3-hMR. Twenty-four hours after transfection, cells were washed once with serum-free DMEM and incubated for 2 h in 500 μL of the same medium containing the indicated concentration of each compound. After this initial period, cells were further incubated with the indicated compounds and 1 × 10^−9^ M [^3^H]-aldosterone (PerkinElmer Life Sciences, Waltham, MA, USA) for one additional hour. Cells were then washed twice with phosphate-buffered saline (PBS: 137 mM NaCl, 2,7 mM KCl, 10 mM Na_2_HPO_4_, 1.8 mM KH_2_PO_4_, pH 7.4) and then extracted with 200 μL of an ethanol:water mix (4:1, v:v). Disintegrations were recorded using liquid scintillation in a β counter. Control conditions included non-transfected cells, transfected cells treated with unlabeled aldosterone or transfected cells, treated with 1 nM [^3^H]-aldosterone in the presence of 1 × 10^−5^ M unlabeled aldosterone. Specific binding was determined by subtracting the number of disintegrations per minute (dpm) in the presence of 1 × 10^−5^ M unlabeled aldosterone from the condition with 1 nM [^3^H]-aldosterone. Each condition was performed in triplicate. Competition of each compound for 1 nM [^3^H]-aldosterone binding was calculated as the percentage of the maximum binding obtained in the absence of compound. Inhibition constants (Ki) were calculated from the percentage competition of 1 × 10^−9^ M [^3^H]-aldosterone binding in the presence of eight difference compound concentrations (in M: 1 × 10^−5^; 5 × 10^−6^; 2.5 × 10^−6^; 1.25 × 10^−6^; 6.25 × 10^−7^; 3.125 × 10^−7^; 1.56 × 10^−7^; 7.8 × 10^−8^). Non-linear regression was performed using Prism 5 (GraphPad) using a model with one binding site and assuming an aldosterone K_d_ of 1 × 10^−9^ M [60], with the following equation:Y = Min + (Max − Min)/(1 + 10^(X − LogKi)^)
where Max and Min are the maximum and minimum values in the Y-axis units and Ki the equilibrium dissociation constant of the unlabeled ligand (in M).

#### 4.3.3. Nuclear Receptor Transcriptional Activity Assays

Receptor transcriptional activity was assessed using a transactivation gene reporter assay, as described [60]. Briefly, COS-7 cells were seeded on 96-well plates in DMEM supplemented with 10% charcoal-stripped FBS (Biowest, Nuaillé, France). Twenty-four hours after seeding, cells were transfected with pcDNA3-hMR, pcDNA3-hGR, or pCMV-FLAG-hAR in combination with pSG5-luc and pSG5-ren in a 5:4:1 proportion, respectively. Twenty-four hours post-transfection, compounds were added at a concentration of 1 × 10^−5^ M. After 2 h of incubation, cells were further treated with agonists (1 × 10^−8^ M aldosterone; 1 × 10^−8^ M testosterone or 0.5 × 10^−6^ M cortisol) in the presence of each compound for 16 h. Assays using hMR included a control condition with 0.25 × 10^−6^ M eplerenone. MR maximum and minimum activity were obtained in each assay using conditions with 1 × 10^−9^ M aldosterone or vehicle (ethanol). Each condition was performed in three parallel wells. After treatments, firefly and *Renilla* luciferase activities were measured sequentially in the cell lysates using a commercially available kit (Dual-Glo; Promega, Madison, WI, USA). Receptor transcriptional activity was calculated as the ratio luc/ren. The activities in each condition were normalized to the activity obtained with vehicle or with agonist as indicated in each figure panel and are expressed a as percentage of the maximum value ± SD.

#### 4.3.4. Ligand-Induced Nuclear Translocation Assays

COS-7 cells were seeded on glass chamber slides divided into 8 culture chambers (0.7 cm^2^/chamber) in DMEM supplemented with 10% charcoal-stripped FBS. Twenty-four hours after seeding, cells were transfected with peGFP-hMR. Twenty-four hours after transfection, cells were treated with each compound at 1 × 10^−5^ M in the presence or absence of 1 × 10^−8^ M aldosterone and incubated for the indicated periods of time. Cells were then washed once with PBS and fixed 4% formaldehyde in PBS for 5 min. After fixation, cells were washed three times with PBS, the walls of the culture chambers were eliminated, and cells were mounted with a coverslip, and mounting medium (25% glycerol, 10% mowiol 4–88, 0.1% 1,4-diazabicyclo [2.2.2]octane -DABCO-, 0.1 M Tris-HCl pH 8.5) supplemented with 0.1 µg/mL 4′,6-diamidino-2-phenylindole (DAPI) to provide nuclear counterstaining. Cell preparations were then examined under a Leica SP8 confocal microscope using a 63x magnification, oil-immersion objective.

#### 4.3.5. Selectivity against a Panel of Nuclear Receptors

Activity on nuclear receptors has been evaluated using a recruitment coactivator, AlphaScreen, assay following the manufacturer’s instructions. Briefly, the assays were performed in white low-volume, 384-well ProxiPlates (Perkin Elmer, Waltham, MA, USA) using a final volume of 15 μL containing the GST-NR-LBD protein and its biotinylated coactivator peptide at specific, previously optimized concentrations, for the determination of a dose-response curve for test compounds over 1 h at room temperature. In the antagonist mode, the stimulation with the compounds was done for 1 h in the presence of a fixed concentration of a reference compound at its EC80 concentration. The plate was read in EnVision instrument (PerkinElmer, Waltham, MA, USA) after a 4 h incubation with 20 μg/mL of donor and acceptor beads. EC_50_ or IC_50_ values were calculated from nonlinear regression curves (GraphPad Prism, San Francisco, CA, USA) from an average of at least two experiments.

## Data Availability

All materials, data, and protocols associated with this publication will be made available to readers upon request.

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
