# Peer review of "Novel 1,4-Dihydropyridine Derivatives as Mineralocorticoid Receptor Antagonists"

_ijms, 2023, doi:10.3390/ijms24032439_

Round 1

Reviewer 1 Report

The authors describe here a comprehensive study aimed at identifying novel 1,4-dihyropyridine (DHP) derivatives acting as mineralocorticoid receptor (MR) antagonists. For this purpose, they first selected a subset of theoretical compounds based on molecular docking studies in the ligand-binding site of MR. Then, they chemically synthesized the most interesting molecules by adapting and improving previously reported methods for 1,4-DHP modification. Finally, they tested the biological activity of these derivatives in three ways: 1) by competition binding assay, displacing labelled aldosterone from MR expressed in transfected COS-7 cells, and measuring their Ki value, 2) by characterizing their agonistic and antagonistic properties using a gene reporter transactivation assay, and 3) by determining their ability to induce fluorescent MR translocation to the cell nucleus. Their selectivity for MR was also demonstrated by showing their inability to stimulate AR or GR or prevent the activation of these receptors by testosterone or cortisol, respectively.

This study is extensively documented and very convincing, but there are unfortunately several mistakes, particularly in the numbering of many figures, making it quite disturbing for the reader. The authors should carefully check all figure numbers within the text.

Specific comments:

1) The R2 chemical group is not defined the same way in the text (p.4, line 153), where it contains the CO moiety, and in the Table S2, where CO is not included.

2) P.11, line 325: “In general, the lower IC50 values when compared with Ki or Kd” should be written “higher IC50 value”

3) P12, line 347: “Fig.6 and supplementary Fig.S2” (not S1)

4) Fig.6: What is the expected effect of combining aldosterone and the various antagonist compounds on the kinetics of MR translocation to the nucleus? Has this experiment been performed?

5) P.13, line 370: “In the antagonist mode …. “ IC50’s (not EC50’s) values should be reported

6) P.13, line 373: Derivative #14 is not shown in Fig. S1

7) P.13, lines 378 and 380: Fig. S1A-B-C-D (not S2A-B-C-D)

8) P.13, line 388: Figure 7 A,B (not Figure 3 A,B)

9) P.13, lines 389-396: Fig.S3 and S4 refer to compound 11 only (not 19 and 20)

10) P.13, line 398: Fig. 7E (not Fig. 3E)

11) Figure 7 legend: Panel E shows compound 20 (yellow) and aldosterone (not compound 19)

12) P.14, line 419: Fig.7F (not 3F) and Fig. S3 and S4 (not S1 and S2)

13) P.14, line 421: Fig. 7E (not 3E)

14) P.15, line 425: Fig. 7F (not 3F)

15) Check references 17, 18, 34

16) Figure S1: Panels A and B are apparently the same (with transfected AR and stimulated by testosterone). Replace Panel A with GR and cortisol data.

17) Figure S2: Nuclear translocation data with compounds 16 and 20 are shown in both Fig.6 and S22, while data with compounds 11 and 17 are not shown in either figure.

Author Response

We would like to thank the reviewer for thoroughly reviewing our manuscript “Novel 1,4-Dihydropyridine derivatives as mineralocorticoid receptor antagonists”, and the valuable comments and recommendations provided. Please, find below a point-by-point answer to your comments.

1) The R2 chemical group is not defined the same way in the text (p.4, line 153), where it contains the CO moiety, and in the Table S2, where CO is not included.

Answer: the R2 group, as the reviewer correctly indicates, is not defined the same way in the main manuscript and in Table S2. This is because in Table S2 the CO group was included within the chemical formula, whereas in Figure 2 it was incorporated within the text. For consistency, we have now modified Table S2 to include the CO group within the table and eliminated it from the formula.

2) P.11, line 325: “In general, the lower IC50 values when compared with Ki or Kd” should be written “higher IC50 value”

A: corrected as indicated.

3) P12, line 347: “Fig.6 and supplementary Fig.S2” (not S1)

A: we thank the reviewer for all the comments regarding the figures. The error arose from a reorganization of the manuscript, we apologize for that. Now, on the supplementary information, Fig. S1 is Fig. S2, and vice versa. 

4) Fig.6: What is the expected effect of combining aldosterone and the various antagonist compounds on the kinetics of MR translocation to the nucleus? Has this experiment been performed?

A: the prediction is that combining the competitive antagonists with aldosterone, MR translocation to the nucleus would be slower, as demonstrated with finerenone and spironolactone (Amazit et al. J Biol Chem 290, 21876-21889, 2015). This is discussed in lines 355-358. To obtained accurate time constants of MR translocation in the presence of each compound with or without aldosterone, we would have to perform multiple experiments and time points, which we have been unable to do, primarily due to the lack of access to automated confocal microscopy facilities.

5) P.13, line 370: “In the antagonist mode …. “ IC50’s (not EC50’s) values should be reported

A: we thank the reviewer for this correction, the text has been changed to “IC50”.

6) P.13, line 373: Derivative #14 is not shown in Fig. S1

A: we thank the reviewer for pointing out this mistake in the text; derivate #14 was not included in the analysis testing selectivity towards GR and AR, shown in current Fig. S2. The text has been corrected to correctly reflect the compounds that were tested in this assay. The purpose of this experiment was not to provide an exhaustive analysis but rather to provide further support to the idea that DHP derivatives generally show a very good selectivity profile towards MR.

7) P.13, lines 378 and 380: Fig. S1A-B-C-D (not S2A-B-C-D)

A: now corrected, see answer to point 3.

8) P.13, line 388: Figure 7 A,B (not Figure 3 A,B)

A: according to the reviewer’s indication, we have corrected the text, which now references the correct figure (Figure 7 instead of Figure 3).

9) P.13, lines 389-396: Fig.S3 and S4 refer to compound 11 only (not 19 and 20)

A: corrected, see answer to point 3. The previous Fig. S5 is now Fig S3 (compound 19), and previous Fig. S6 is now Fig. S4 (compound 20).

10) P.13, line 398: Fig. 7E (not Fig. 3E)                                   

A: corrected.

11) Figure 7 legend: Panel E shows compound 20 (yellow) and aldosterone (not compound 19)

A: We thank the reviewer for this observation. Now it is correctly indicated as compound 20.

12) P.14, line 419: Fig.7F (not 3F) and Fig. S3 and S4 (not S1 and S2)

A: this has now been corrected (please see answers to points 3 and 8). Now the reference to Fig. 7 is correct in the text and figures for compound 11 in the supplementary information (Fig. S5 and S6) also match the text.

13) P.14, line 421: Fig. 7E (not 3E)                    

A: corrected.

14) P.15, line 425: Fig. 7F (not 3F)

A: corrected.

15) Check references 17, 18, 34

A: The references have been corrected as needed. Please note that numbers have changed due to the inclusion of a new reference, as suggested by reviewer 2.

16) Figure S1: Panels A and B are apparently the same (with transfected AR and stimulated by testosterone). Replace Panel A with GR and cortisol data.

A: we thank the reviewer for spotting this mistake; indeed, panel A and B had accidentally repeated data from cells transfected with AR and stimulated with T. It has now been corrected to show the corresponding data with GR and stimulation with CORT in panel A and AR/T in panel B. Please note that Figure S1 is now S2, as pointed out above.

17) Figure S2: Nuclear translocation data with compounds 16 and 20 are shown in both Fig.6 and S22, while data with compounds 11 and 17 are not shown in either figure.

A: thank you for identifying this mistake, due to using an earlier version of the figure. Fig. S1 (previous Fig. S2) now shows compounds 11, 12a, 13ab, 15 and 17.

Reviewer 2 Report

The Authors report the identification of a novel series of 1,4-DHP that has been guided by structure-based drug design, focusing on the less explored DHP position 2. Interestingly, substituents at this position might interfere with MR helix H12 disposition, which is essential for the recruitment of co-regulators. Several of the newly synthesized 1,4-DHPs show interesting properties, as MRAs, and have a good selectivity profile. These 1,4-DHPs promote MR nuclear translocation with less efficiency than the natural agonist aldosterone, which explains, at least in part, its antagonist character. Molecular dynamic studies are suggestive of several derivatives interfering with the disposition of H12 in the agonist-associated conformation, and thus, they might stabilize an MR conformation unable to recruit co-activators.

The study is interesting. The authors conclude that the new compounds have only partial antagonistic activity unlike spironolactones
Comments:

-Introduction:it is not possible that the new compounds are able to prevent frequent hyperkalemia during therapy with spironolactone and derivatives, otherwise they would not be antagonists
-Biological activity would have to be evaluated in vivo, for example, in adrenalectomized rats treated with aldosterone alone or with addition of varying doses of the new compounds by assessing urinary sodium and potassium excretion.
-The authors must also assess possible affinity for the glucocorticoid receptor, as reported for    aldosterone, or explain why the compounds have only an affinity for the mineralocorticoid receptor.
In vitro and in vivo toxicity studies are desirable at the doses necessary to exert receptor biological action
-Report how cyproheptadine as a partial non-steroidal antagonist for the aldosterne receptor Clin Endocrinol Metab. 1983 Feb;56(2):397-400. doi:10.1210/jcem-56-2-397 has also been studied in the literature.

Author Response

We would like to thank the reviewer for thoroughly reviewing our manuscript “Novel 1,4-Dihydropyridine derivatives as mineralocorticoid receptor antagonists”, and the valuable comments and recommendations provided. Please, find below a point-by-point answer to your comments.

- Introduction: it is not possible that the new compounds are able to prevent frequent hyperkalemia during therapy with spironolactone and derivatives, otherwise they would not be antagonists

Answer: we agree with the reviewer; in the Introduction, we describe the evolution of MR pharmacology and the attractive possibility of finding modulatory compounds that are not pure inhibitors. We do not make the claim that the new compounds presented in this work have these characteristics. Our data indicates that they are competitive antagonists of the receptor, as described in the Conclusions section.  

- Biological activity would have to be evaluated in vivo, for example, in adrenalectomized rats treated with aldosterone alone or with addition of varying doses of the new compounds by assessing urinary sodium and potassium excretion.

A: we agree that this would certainly be the next logical step in our project. However, we feel that this would be out of the scope of the present manuscript, which mainly focuses on exploring one particular position in the 1,4-DHP scaffold (position 2) to produce different derivatives that bind MR and inhibit it.

- The authors must also assess possible affinity for the glucocorticoid receptor, as reported for    aldosterone, or explain why the compounds have only an affinity for the mineralocorticoid receptor.

A: we thank the referee for this comment. Now, we have further explained the selectivity for MR in comparison with other steroid receptors (see page 13).

- In vitro and in vivo toxicity studies are desirable at the doses necessary to exert receptor biological action.

A: we agree with the reviewer in that toxicity studies are important to progress in the development of new MRAs. However, this study mainly focuses on the molecular properties of substituents at position 2 of the 1,4-DHP scaffold, as mentioned above. Further developing these compounds will surely need performing toxicity studies in the future.

- Report how cyproheptadine as a partial non-steroidal antagonist for the aldosterne receptor Clin Endocrinol Metab. 1983 Feb;56(2):397-400. doi:10.1210/jcem-56-2-397 has also been studied in the literature.

A: we thank the reviewer for pointing out at this study, which is now mentioned in the Introduction (lines 67-68).